# Toughened Poly(lactic acid)/BEP Composites with Good Biodegradability and Cytocompatibility

**DOI:** 10.3390/polym11091413

**Published:** 2019-08-28

**Authors:** Qingguo Wang, Yongxuan Li, Xue Zhou, Tongyao Wang, Liyan Qiu, Yuanchun Gu, Jiabing Chang

**Affiliations:** 1Key Laboratory of Rubber-Plastics of Ministry of Education, Qingdao University of Science and Technology, Qingdao 266042, China; 2Shandong Provincial Key Laboratory of Rubber-Plastics, Qingdao 266042, China

**Keywords:** PLA, BEP, composites, biodegradability, cytocompatibility, toughness

## Abstract

Using novel biodegradable elastomer particles (BEP) prepared by the technologies of melt polycondensation, emulsification, and irradiation vulcanization, we successfully prepared advanced poly(lactic acid) (PLA)/BEP composites with higher toughness, higher biodegradability, and better cytocompatibility than neat PLA by means of the melt blending technology. The experimental results revealed that the elongation at break of the PLA/BEP composites containing 8 parts per hundred rubber (phr) by weight BEP increased dramatically from 2.9% of neat PLA to 67.1%, and the notched impact strength increased from 3.01 to 7.24 kJ/m^2^. Meanwhile, the biodegradation rate of the PLA/BEP composites increased dramatically in both soil environment and lipase solution, and the crystallization rate and crystallinity of the PLA/BEP composites increased significantly compared to those of neat PLA. The methyl thiazolyl tetrazolium (MTT) assay also showed that the viability of L929 cells in the presence of extracts of PLA/BEP composites was more than 75%, indicating that the PLA/BEP composites were not cytotoxic and had better cytocompatibility than neat PLA. Research on advanced PLA/BEP composites opens up new potential avenues for preparing advanced PLA products, especially for advanced biomedical materials.

## 1. Introduction

Poly(lactic acid) (PLA), a bio-based biodegradable thermoplastic aliphatic polyester derived from biomass resources such as corn and sugarcane [1], has attracted a lot of attention from researchers interested in environmentally friendly materials, especially in biomedical materials. PLA has been developed for many different applications in biomedicine, including bioabsorbable screws [2,3], tissue engineering [4], stents [5,6,7,8], dental materials [9], resorbable sutures [10,11] and drug delivery systems [12,13], because of its biocompatibility, biodegradability, low immunogenicity, high strength and modulus [14,15,16]. However, its excessive brittleness and slow crystallization rate and biodegradation rate severely limit its further applications, especially for some specific products which require high toughness, short manufacture time, and good biocompatibility [17,18,19].

Therefore, some techniques have been developed to improve the performance and further applications of PLA. For instance, blending with vulcanized elastomers (such as poly (styrene-butadiene-styrene) (SBS) [20], nitrile-butadiene rubber (NBR) [21,22], thermoplastic elastomer (TPE) [23], and polyurethane (PU) [24]), nanoscale inorganic fillers (such as graphene [25,26], carbon nanotubes [27], clay [28,29], fibers [30,31]), and plasticizers (such as polyethylene glycol (PEG) [32], polypropylene glycol (PPG) [33], and acetyl tributyl citrate (ATBC) [34]) is commonly considered to improve the toughness and crystallization rate of PLA. Unfortunately, vulcanized elastomers have poor dispersion in a PLA matrix and have poor interface compatibility with it, decreasing the crystallization property of PLA. Inorganic fillers can enhance the stiffness obviously, although they only marginally improve the toughness or even decrease it. Low-molecular-weight plasticizers toughen PLA, but this is obtained at the cost of losing stiffness. Meanwhile, the above vulcanized elastomers, inorganic fillers, and plasticizers are not biodegradable and can deteriorate the valuable biodegradability and biocompatibility of PLA. Therefore, it is very important to find the suitable materials which can increase not only the toughness of PLA but also its crystallization rate, while not sacrificing its valuable biodegradability and biocompatibility. 

In our previous work, using some bio-based monomers, such as propanediol, butanediol, succinic acid, sebacic acid, itaconic acid, and fumaric acid, we successfully fabricated novel biodegradable elastomer particles (BEP) by technologies based on melt polycondensation of aliphatic unsaturated polyester, polyester emulsified, and electron beam irradiation vulcanization [35]. It is worth noting that micro-nanoscale BEP are biodegradable due to the ester group in the macromolecular chains of the aliphatic polyester. Also, BEP has a low glass transition temperature of approximately −50 °C, controllable crosslinking density, and good compatibility with PLA, which makes it possible that BEP can be dispersed well in a PLA matrix, imparting some excellent properties to the PLA.

In this work, the novel BEP were used for the first time to prepare PLA/BEP composites. This paper focuses on the effects of BEP on toughness, crystallization, biodegradation, and cytocompatibility of PLA/BEP composites in order to obtain some advanced PLA composites that can be used for biomedical materials, food packaging, 3D printing, and more environmentally friendly products.

## 2. Materials and Methods 

### 2.1. Materials

BEP with particle size of about 200 nm and gel content of 85% were fabricated in Qingdao University of Science and Technology (Qingdao, China); PLA (4032D) was supplied by NatureWorks LLC. (Minnetonka, MN, USA); hydrolytic stabilizer (HMV-8CA) was supplied by Nisshinbo Industries Inc. (Tokyo, Japan); lipase, 20000 U/g, was supplied by Xinyu Food Additives Co., Ltd. (Wuxi, China); mixed phosphate (pH buffer), was supplied by Shanghai INESA Scientific Instrument Co., Ltd. (Shanghai, China); the mouse fibroblastic L929 cells was supplied by the type Culture Collection of the Chinese Academy of Sciences (Shanghai, China); Dulbecco′s modified eagle′s medium (DMEM) was supplied by Thermo Fisher Scientific Inc. (Waltham, MA, USA); fetal bovine serum (FBS), endotoxin level ≤10 EU/mL, was supplied by Thermo Fisher Scientific Inc. (Waltham, MA, USA); penicillin-streptomycin solution (penicillin: 10000 units/mL, streptomycin: 10 mg/mL) was supplied by Biological Industries (Beit Haemek, Israel); methyl thiazolyl tetrazolium (MTT), purity 98%, was supplied by Sinopharm Chemical Reagent Co., Ltd. (Shanghai, China); dimethyl sulfoxide (DMSO), analytical reagent (AR), was supplied by Sinopharm Chemical Reagent Co., Ltd. (Shanghai, China).

### 2.2. Preparation of PLA/BEP Composites

Firstly, PLA and BEP were placed in a vacuum-drying oven at 80 °C for 2 h, then PLA, BEP, and the hydrolytic stabilizer (HMV-8CA) (listed in Table 1) were mixed in a torque rheometer (RM-200C, Harbin Hapro Electric Technology Co., Ltd., Harbin, China) at a temperature of 175 °C for 13 min and a rotor speed of 50 r/min. PLA and PLA/BEP composite test specimens were prepared with the help of a mini-injection molder.

### 2.3. Characterization

#### 2.3.1. Mechanical Properties

Tensile tests of the PLA and PLA/BEP composites were performed using a universal testing machine (UTM; GT-TCS-2000, Gotech Testing Machines Inc., Taichung, China) according to ISO 527-2-2012 at room temperature. The specimens were shaped with total length of 75 mm, gauge length of 20 mm, and thickness of 2 mm. The crosshead speed was 5 mm/min. An notched Izod impact test was performed using a pendulum impact tester (GT-7045-MDH, Gotech Testing Machines Inc., Taichung, China) according to ISO 180-2000 at room temperature. The size of the impact test specimen was 80 mm × 10 mm × 4 mm, and the radius of the notch base was 0.25 ± 0.05 mm.

#### 2.3.2. Surface Morphology of the Tensile Specimens and the Notched Impact Specimens

The neck region surface of the tensile specimens and the fracture surface of the tensile specimens and notched impact specimens were observed on a scanning electron microscope (SEM; S-4800, Hitachi Ltd., Tokyo, Japan), and the surfaces were sputter-coated with a thin gold layer before observation.

#### 2.3.3. Crystallization Behavior and Isothermal Crystallization Kinetics

An isothermal crystallization test was performed by a differential scanning calorimeter (DSC; 2041F1, Netzsch Group, Selb, Germany). Samples of about 10 mg of PLA or PLA/BEP composites were heated from room temperature to 200 °C at a rate of 50 °C/min and kept at 200 °C for 3 min to eliminate their thermal history. Then, the samples were rapidly cooled to 115 °C. After the crystallization process was completed, the measurement was performed during a second heating from 115 to 200 °C at a rate of 10 °C/min. The *T*_m_, Δ*H*_m_, and *t*_m_ were calculated from the DSC melting thermograms, and the crystallinity of PLA and PLA/BEP composites was calculated by Equation (1), where Δ*H*_0_ is the fusion enthalpy of perfectly crystallized PLA:Crystallinity (%) = Δ*H*_m_/Δ*H*_0_ × 100%(1)

The heat flow data of isothermal crystallization DSC curves were converted into a fraction relative to the final crystallinity level according to the Avrami kinetic model (Equation (2)):(2)$$X_{t}=1-\exp\left(-Kt^{n}\right)$$ where *X*_t_ is the relative crystallinity, *K* is the crystallization rate constant, and *n* is the Avrami exponent. In order to estimate the values of *n* and *K*, the double-logarithmic Equation (3) was employed to analyze the crystallization kinetics of the PLA and PLA/BEP composites.
ln[−ln(1 − *X*_t_)] = *n*ln*t* + ln*K*(3)

The half time of crystallization (*t*_1/2_) is defined as the time required to attain the *X*_t_ = 0.5, which is calculated by Equation (4):
(4)$$t_{1/2}={\left(\frac{\ln2}{K}\right)}^{\frac{1}{n}}$$

#### 2.3.4. Crystallization Morphology

Firstly, PLA and PLA/BEP composite specimens were prepared on glass slides at 185 °C. Then, they were heated from room temperature to 200 °C at a heating rate of 50 °C/min and kept at 200 °C for 3 min. Finally, the specimens were cooled from 200 to 115 °C at a rate of 100 °C/min, and their crystallization morphology was observed on a polarizing microscope (POM; BX51, Olympus Corporation, Tokyo, Japan).

#### 2.3.5. X-ray Diffraction Test

X-ray diffraction patterns of PLA and PLA/BEP composite films annealed at 115 °C for 90 min were obtained in the reflection mode by using an X-ray diffractometer (XRD; Ultima IV, Rigaku Corporation, Tokyo, Japan) with Cu-Kα radiation wavelength (λ = 1.54 Å), 40 kV, 40 mA. Scattered radiation was detected in the angular range of 10°–30° (2θ) at a scanning rate of 5 °/min.

#### 2.3.6. Biodegradation Test

##### Biodegradation Test in Soil

The dried PLA and PLA/BEP composite samples with size of 15 mm × 15 mm × 1 mm were weighed, and their corresponding weights were denoted as *W*_0_; then, they were buried in soil. Four months later, the above samples were cleaned and dried in a vacuum-drying oven to a constant weight, which was denoted as *W*_t_. Finally, the mass loss rate was calculated using Equation (5):Mass loss rate (100%) = (*W*_0_ − *W*_t_)/*W*_0_ × 100%(5)

##### Biodegradation Test in Lipase Solution

The dried PLA and PLA/BEP composite samples with size of 10 mm × 10 mm × 1 mm were weighed, and their corresponding weights were denoted as *W*_0_; then, they were placed into separate sampling tubes, and 10 mL of a lipase solution (5 mg/mL, pH 6.8) was added into each sampling tube at the temperature of 37 °C. The samples were incubated for 10 days. Finally, the samples in lipase solution were cleaned and dried in a vacuum-drying oven to a constant weight, which was denoted as *W*_t_, and the mass loss rate was calculated using Equation (5) too.

#### 2.3.7. Surface Morphology of PLA and PLA/BEP Composites after Biodegradation

The surface morphology of various biodegraded PLA and PLA/BEP composite samples was observed by a stereo microscope (SMZ1500, Nikon Corporation, Tokyo, Japan).

#### 2.3.8. Cytocompatibility Analysis

##### Cell Culture

The mouse fibroblastic L929 cells were cultured in a humidified incubator with 5% CO_2_ at 37 °C using DMEM containing 10% FBS, 100 U/mL penicillin, and 100 μg/mL streptomycin. For the MTT assay, the cells were seeded in 96-well culture plates at a density of 5000 cells per well. Then, 100 μL of extracts of PLA and PLA/BEP composites at different concentrations (0.25, 0.5, 1, 2, 5 mg/mL) was added to the culture medium. Meanwhile, 100 μL phenol (0.5%) was used as a positive control, and 100 μL phosphate-buffered saline (PBS, pH 7.4) was used as a negative control. Non-treated control cells (blank) were analyzed to compare the growth inhibition. The entire plate was observed in an inverted microscope (CKX41, Olympus Corporation, Tokyo, Japan) after 24, 48, and 72 h of incubation.

##### MTT Assay

After 24, 48 and 72 h of incubation with extracts medium, 10 μL of MTT solution with 5 mg/mL was added to each well, and the plates were incubated at 37 °C in 5% CO_2_/air for 4 h. Next, the medium was removed, and 150 μL of DMSO was added to each wells. The optical density (OD) was measured in an absorbance microplate reader (ELx800, BioTek Instruments Inc., Winooski, VT, USA) at 490 nm. All experiments were repeated three times. Finally, cell viability was calculated using the Equation (6):
(6)$$\mathrm{Cell}\;\mathrm{viability}=\frac{{\mathrm{OD}}_{\mathrm{experiment}}-{\mathrm{OD}}_{\mathrm{blank}}}{{\mathrm{OD}}_{\mathrm{negative}}-{\mathrm{OD}}_{\mathrm{blank}}}$$

## 3. Results and Discussion

### 3.1. Effects of BEP on the Toughness of PLA/BEP Composites

Figure 1 presents the relationship between BEP content and the toughness of the PLA/BEP composites. With the increase of BEP concentration, the impact strength and elongation at break of the PLA/BEP composites increased significantly. The elongation at break of the PLA/BEP composites containing 8 phr BEP increased dramatically from 2.9% of the neat PLA to 67.1%, and the notched impact strength increased from 3.01 to 7.24 kJ/m^2^. These phenomena can be explained by the multiple crazing theory [36]. On the one hand, the well-dispersed BEP in the PLA matrix have an effect on stress concentration during the tensile process, inducing the initiation and extension of crazing in the PLA matrix. Both crazing initiation and crazing extension can consume energy. Accordingly, the polymer absorbs a high amount of energy and avoids a highly localized strain process, which leads to higher elongation at break and impact strength [37,38]. On the other hand, crazing extension will terminate when they meet with some other BEPs, thereby preventing the generation of cracking in the PLA matrix and increasing the toughness of the PLA/BEP composites [39]. However, when BEP loading increases to 10 phr, elongation at break and impact strength of the PLA/BEP composites slowly decrease because excessive BEP tend to aggregate in the PLA matrix, thus weakening crazing initiation and crazing extension in the PLA matrix.

Figure 2 shows some SEM micrographs of the impact fracture surfaces of neat PLA and PLA-8 composites. In Figure 2a, the impact fracture surface of neat PLA is smooth, and there is no plastic deformation, indicating that a brittle fracture process occurred in neat PLA. However, there are some voids and a certain roughness on the impact fracture surface of the PLA-8 composites (Figure 2b), which indicates that a ductile fracture process occurred in the PLA-8 composites. The above results show that BEP dispersed in the PLA matrix can absorb the impact energy and prevent further fractures in the PLA matrix.

Figure 3 shows some SEM micrographs of the tensile neck region surface of neat PLA and PLA/BEP composites. In Figure 3a, the tensile neck region surface of neat PLA is smooth, whereas the tensile neck region surface of the PLA/BEP composites exhibits the stress whitening phenomenon (in Figure 3b,c), which indicates that the crack propagation paths were highly bifurcated and crack propagation absorbed a considerable amount of strain energy before the PLA matrix was completely destroyed.

Figure 4 shows some SEM micrographs of tensile fracture surface of neat PLA and PLA/BEP composite specimens, indicating a large plastic deformation of the PLA matrix in the PLA/BEP composites, while the fracture surface of neat PLA is smooth, which confirms the above conclusions that BEP in the PLA matrix have the effect of absorbing energy and causing a ductile fracture process in the PLA matrix.

On the basis of the above SEM micrographs, we inferred that BEP toughen PLA by forming large interfaces; strong interfacial interactions between the PLA matrix and BEP deform the PLA matrix and increase the absorption of energy until the PLA matrix breaks. Thus, the impact strength and elongation at break of the PLA/BEP composites increase considerably with increasing BEP loading.

### 3.2. Effects of BEP on the Crystallization Properties of PLA/BEP Composites

The isothermal crystallization kinetic parameters of neat PLA and PLA/BEP composites are listed in Table 2. As shown in Table 2, the *t*_1/2_ of PLA-8 is 41% faster than that of neat PLA, while the crystallinity increases of 50.6%, which means that BEP can accelerate the crystallization rate and increase the crystallinity of PLA. The *n* values of the PLA/BEP composites are higher than that of neat PLA, indicating that the PLA/BEP composites present more extensive three-dimensional crystal growth than neat PLA during the crystallization process. The *K* value of PLA-8 also increases of 59.6% compared to that of neat PLA. Therefore, BEP act as an effective nucleating agent and accelerate the crystallization rate of PLA by providing nucleation sites and facilitating crystal growth.

Figure 5 shows the melting DSC thermograms curves of neat PLA and PLA/BEP composites. Interestingly, the PLA/BEP composites have two melting peaks at about 160 and 167 °C, while the neat PLA has only one melting peak at 165.4 °C, which indicates that two kinds of crystal forms appeared in the PLA/BEP composites crystallization process [40]. As is well known, the α crystal form of PLA, which grows during melting or cold crystallization, is its most common and stable crystal form. Zhang et al. found that the α′ crystal form coexists with the α crystal form when *T*_c_ is between 100 and 120 °C [41]. It can be speculated that BEP in the PLA matrix accelerated the formation of the α′ crystal form and that 160 °C is the melting point of the α′ crystal form, while 167 °C is the melting point of the α crystal form. 

Figure 6 shows some XRD patterns of neat PLA and PLA/BEP composites. Two strong diffraction peaks appeared at 16.92° and 19.22° in the XRD patterns of neat PLA, which correspond to two kinds of diffractions of the (200)/(110) crystal plane and (203) crystal plane, respectively [42]. However, the diffraction peaks of the (200)/(110) crystal plane in the XRD patterns of PLA-4 and PLA-8 composites shifted to low 2θ, from 16.92° to 16.82° and 16.76°, respectively. These shifts were observed during the formation of the α and α′ crystal phases in the range from 100 to 120 °C. In addition, BEP in the PLA matrix increased the intensity of the XRD patterns, indicating that BEP had the effect of improving the crystallinity of PLA. 

Figure 7 presents some polarizing optical micrographs of neat PLA and PLA-8 composites with various isothermal crystallization times at 115 °C. It can be seen that no crystal of neat PLA (Figure 7a) was observed during 5 min of isothermal crystallization, and a small number of crystals grew at a slow crystallization rate after 10 min of isothermal crystallization. Meanwhile, the number of those crystals did not increase, whereas the size of the spherulites increased after 15 min of isothermal crystallization. However, many small spherulites of PLA-8 composites with a morphology different from that of neat PLA grew after 5 min of crystallization, and the spherulites radius of PLA-8 was smaller than that of neat PLA (Figure 7b), which possibly can be explained by the fact that the twisting of lamellae resulted in varied crystalline structures of PLA [43]. After 10 min of isothermal crystallization, the number of spherulites of PLA-8 increased rapidly, and after 15 min, the spherulites of PLA-8 nearly covered the entire field. It can be inferred that BEP act as a nucleating agent in the PLA crystallization process and accelerate PLA crystallization rate.

### 3.3. Effects of BEP on the Biodegradation of PLA/BEP Composites

In general, the soil burying test and composting test are performed to evaluate the biodegradability of plastics in natural environments [44,45]. The degradation of PLA in soil or compost occurs through the following process. During the initial phases of degradation, PLA chains with high molecular weight are hydrolyzed to form chains with lower molecular weight, and this reaction can be accelerated by acids or bases and is also affected by temperature and moisture [46]. Also, some microorganisms in the compost catalyze the degradation, probably through the hydrolytic scission of ester groups into an acid and an alcohol. This process enables the conversion of the chains with low molecular weights to CO_2_, water, and humus.

Figure 8 shows some stereo microscope images of neat PLA and PLA/BEP composites degraded in soil for 4 months. The surface of neat PLA after degradation remained smooth and clean. However, extensive mildew was observed on the surfaces of the PLA/BEP composites, among which the PLA-8 sample had the most extensive mildew growth, indicating that the amount of mildew increased significantly on the PLA/BEP composites with increasing BEP loading.

Figure 9 presents the mass loss of neat PLA and PLA/BEP composites after 4 months in soil. With increasing BEP loading, the mass loss of the PLA/BEP composites markedly increased, and the mass loss of the PLA-10 increased 3.6 times (from 0.31 % to 1.44 %). 

Pranamuda et al. found that microorganisms which can degrade PLA are rare in soil, and only one strain of such microorganisms (*Amycolatopsis sp*.) was isolated [47]. Therefore, neat PLA has a low degradation rate in soil. However, the PLA/BEP composites presented a better degradability than neat PLA, which indicates that BEP exhibit good degradability and produce some low-molecular-weight aliphatic acids and aliphatic alcohols in the degradation process [35]. Thereby, the presence of BEP in the PLA matrix increase the degradation rates of the PLA/BEP composites.

Figure 10 presents the mass loss of neat PLA and PLA/BEP composites degraded in a lipase solution for 10 days. With the increase of BEP loading, the mass loss of the PLA/BEP composites significantly increased, and the mass loss of PLA-10 increased 1.46 times (from 1.3% to 3.2%) compared to that of neat PLA. In order to explain this phenomenon, we present a degradation schematic of the PLA/BEP composites in lipase solution (Figure 11). The large specific surface area of BEP can increase the contact area between the PLA/BEP composites and the lipase solution, making the PLA/BEP composites more susceptible to biodegrade by hydrolysis and enzymatic activity. Meanwhile, BEP are readily exposed to enzymes, and enzymes can easily break apart the ester bonds in BEP. Therefore, BEP easily degrade into acids and alcohols. As a catalyst, the H^+^ in the acid accelerates the degradation process of the PLA matrix, and the voids and gaps formed after BEP degradation can also facilitate the penetration of the lipase solution into the PLA matrix [48]. Thus, the degradation rates of the PLA/PEP composites increases.

### 3.4. Effects of BEP on the Cytocompatibility of PLA/BEP Composites

Figure 12 presents the results of the MTT assay which was conducted to check cell viability after 24, 48, and 72 h of incubation with different concentrations of extracts. Cell viability in the presence of extracts of the PLA/BEP composites was higher than in the presence of extracts of neat PLA (more than 75% in the presence of all the PLA/BEP composite extracts). At the time point of 72 h, the viability of cells cultured in the presence of all extracts of the PLA/BEP composites was higher than that measured at 24 h for the corresponding extracts. Figure 13 presents microscopic images of the control groups and the experiment groups treated with 2 mg/mL of extracts of the PLA/BEP composites. The cell morphologies were determined on the basis of cell distribution and mean cell aspect ratio, with L929 cells having an aspect ratio greater than 1.5 [49]. The addition of extracts did not cause considerable changes in the morphology of the cells, such as rounding or shrinking, and all the images are more or less identical to that of the negative control. So, it can be concluded that the PLA/BEP composite extracts at the tested concentration do not exert cytotoxic effects on cell proliferation, and the PLA/BEP composites have better cytocompatibility than PLA, which makes them a potential material for biomedicine. 

## 4. Conclusions

Using micro-nanoscale BEP prepared by melt polycondensation, emulsification, and irradiation vulcanization, we successfully prepared advanced PLA/BEP composites by a simple melt blending technology. Elongation at break of the PLA/BEP composites containing 8 phr BEP increased dramatically from 2.9% of neat PLA to 67.1%, and the notched impact strength increased from 3.01 to 7.24 kJ/m^2^. The crystallization rate and crystallinity of the PLA/BEP composites were higher than those of neat PLA. Meanwhile, two kinds of crystal forms were formed in the PLA/BEP composites crystallization process. It is worth noting that BEP had the effect of improving the biodegradation rate of the PLA/BEP composites in both soil environment and lipase solution. Also, the PLA/BEP composites showed better cytocompatibility. This research is very important for preparing some advanced PLA products which meet the requirements of high toughness, high biodegradability, good cytocompatibility, and short manufacturing time.

## Figures and Tables

**Figure 1 polymers-11-01413-f001:**
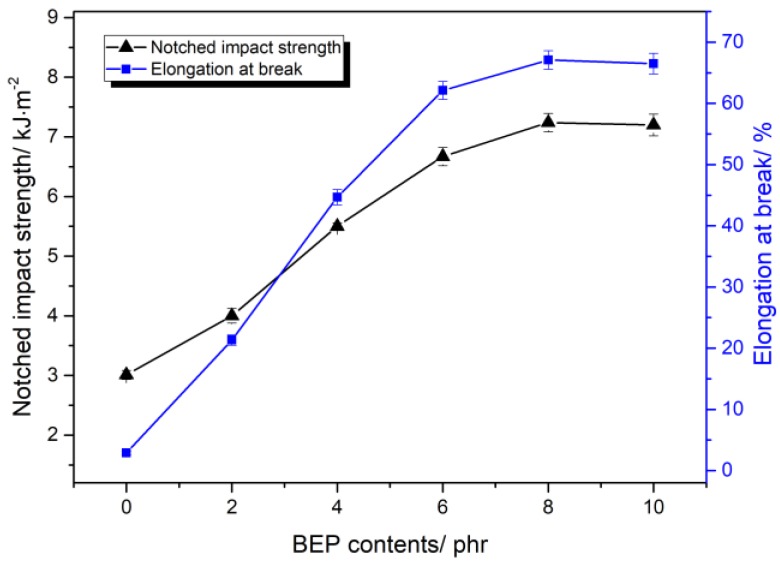
Impact strength and elongation at break of PLA/BEP composites with various BEP amounts.

**Figure 2 polymers-11-01413-f002:**
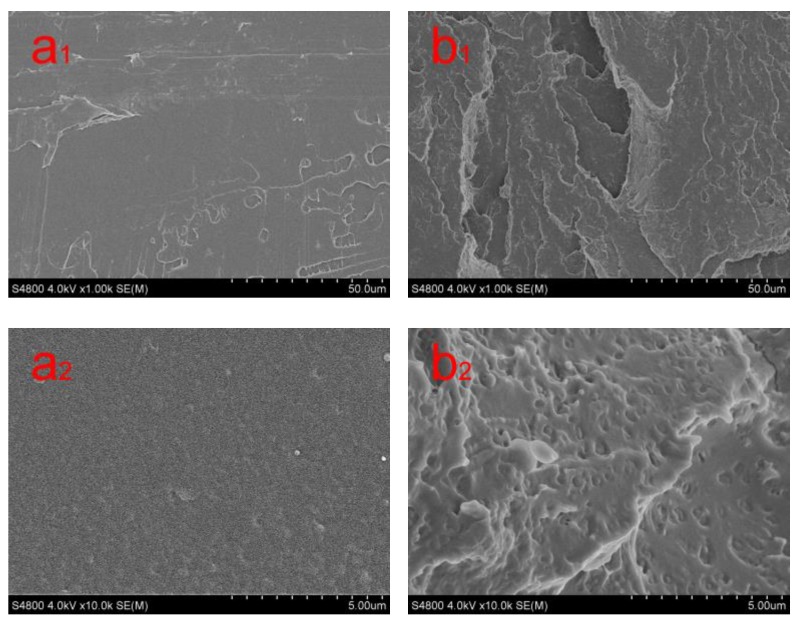
SEM micrographs of the impact fracture surface of neat PLA and PLA-8 composites, (**a_1_**) PLA (1k magnification); (**b_1_**) PLA-8 (1k magnification); (**a_2_**) PLA (10k magnification); (**b_2_**) PLA-8 (10k magnification).

**Figure 3 polymers-11-01413-f003:**

SEM micrographs of the neck regions of neat PLA and PLA/BEP composites at 1k magnification, (**a**) PLA; (**b**) PLA-4; (**c**) PLA-8.

**Figure 4 polymers-11-01413-f004:**
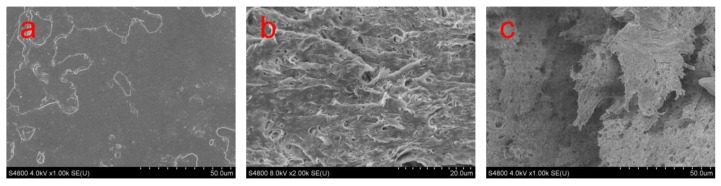
SEM micrographs of the tensile fracture surface of neat PLA and PLA/BEP composites at 2k magnification, (**a**) PLA; (**b**) PLA-4; (**c**) PLA-8.

**Figure 5 polymers-11-01413-f005:**
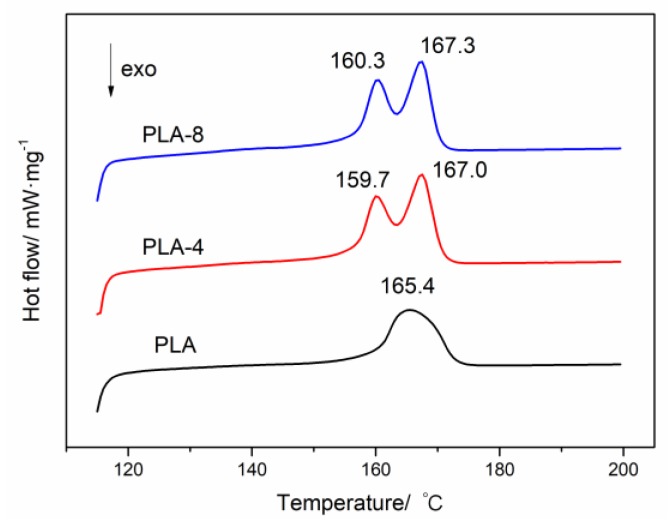
Melting DSC thermograms curves of neat PLA and PLA/BEP composites.

**Figure 6 polymers-11-01413-f006:**
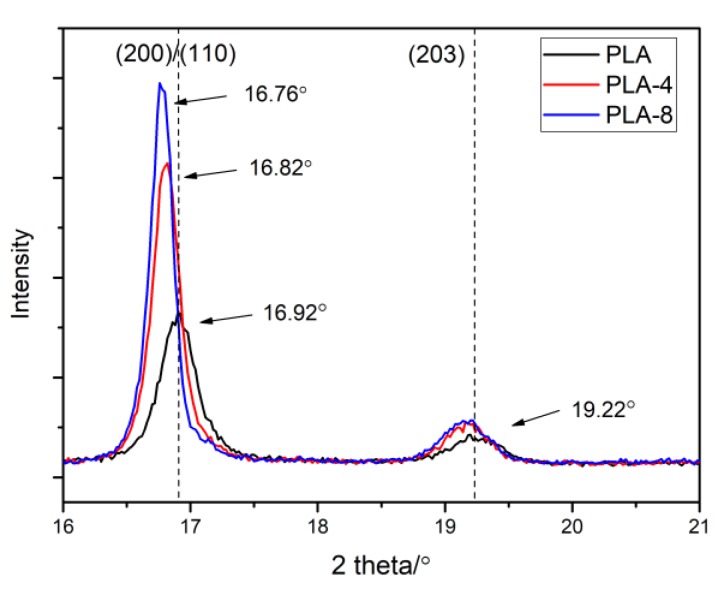
XRD patterns of neat PLA and PLA/BEP composite films annealed at 115 °C for 90 min.

**Figure 7 polymers-11-01413-f007:**
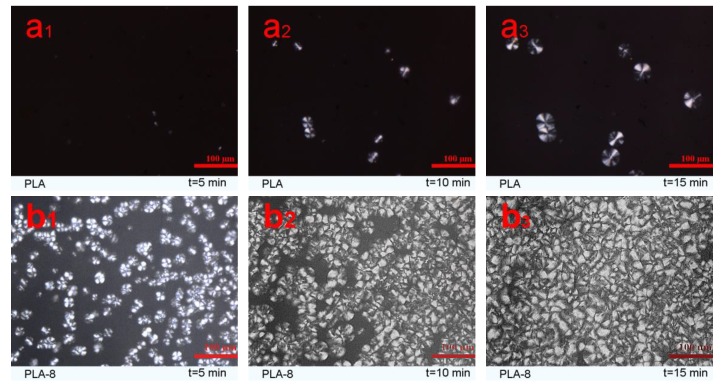
Polarizing optical micrographs of the spherulitic morphology of neat PLA and PLA-8 composites, (**a**) PLA; (**b**) PLA-8.

**Figure 8 polymers-11-01413-f008:**
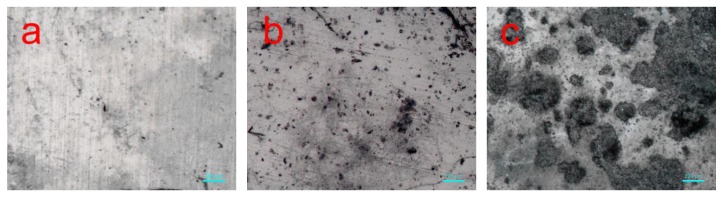
Stereo microscope photos of neat PLA and PLA/BEP composites after 4 months of degradation in soil, (**a**) PLA; (**b**) PLA-4; (**c**) PLA-8.

**Figure 9 polymers-11-01413-f009:**
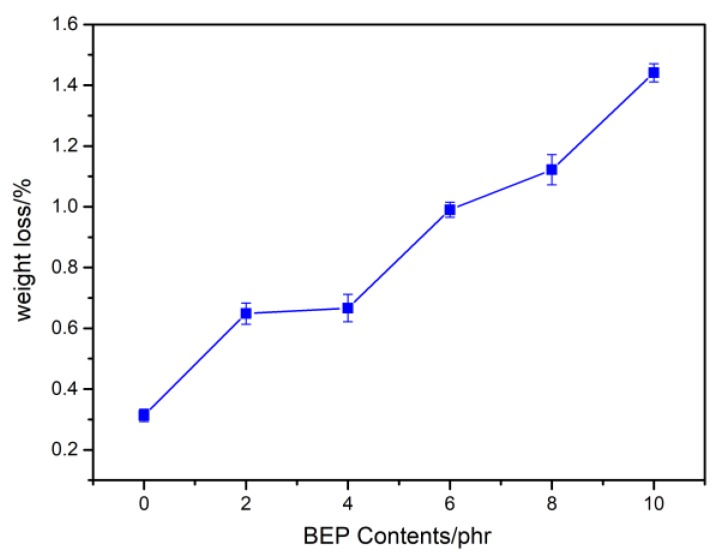
Mass loss of neat PLA and PLA/BEP composites after degradation in soil for 4 months.

**Figure 10 polymers-11-01413-f010:**
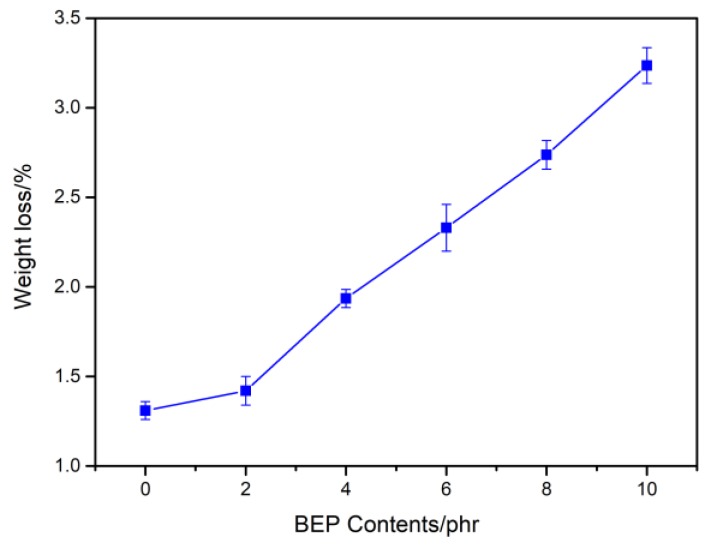
Mass loss of neat PLA and PLA/BEP composites after degradation in lipase solution for 10 days.

**Figure 11 polymers-11-01413-f011:**
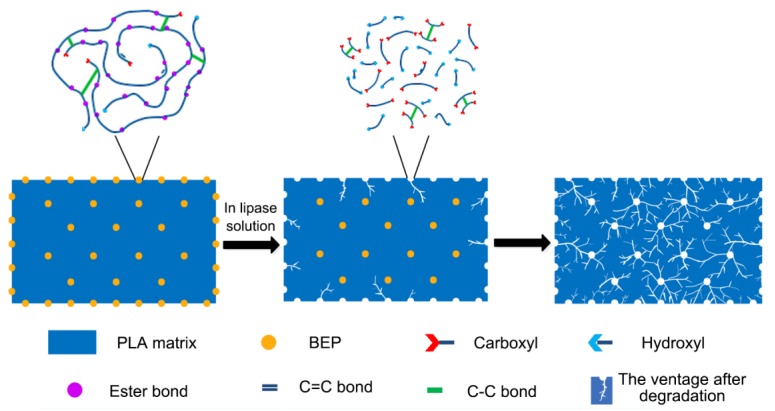
Degradation schematic of the PLA/BEP composites in a lipase solution.

**Figure 12 polymers-11-01413-f012:**
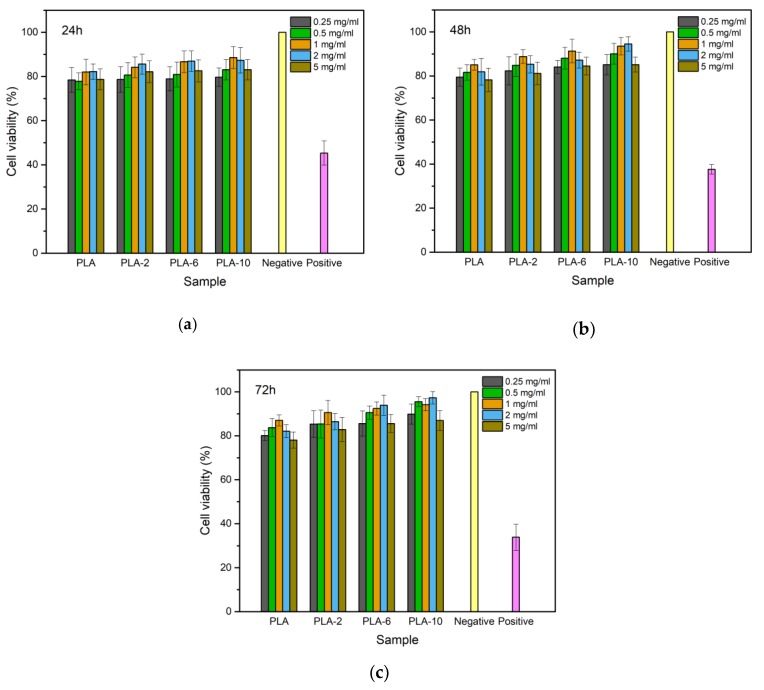
Cell viability of L929 cells in the presence of extracts of neat PLA and PLA/BEP composites at different concentrations and incubation times: (**a**) 24 h; (**b**) 48 h; (**c**) 72 h.

**Figure 13 polymers-11-01413-f013:**
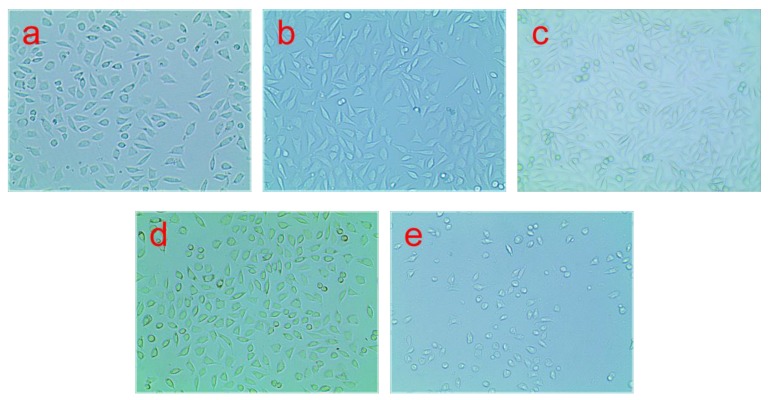
Micrograph of L929 cells at 100 magnification: (**a**) 24 h PLA-10; (**b**) 48 h PLA-10; (**c**) 72 h PLA-10; (**d**) negative control, after 24 h; (**e**) positive control, after 24 h.

**Table 1 polymers-11-01413-t001:** Composition of various poly(lactic acid) (PLA)/biodegradable elastomer particles (BEP) composites. Phr: parts per hundred rubber.

Sample Numbers	PLA	PLA-2	PLA-4	PLA-6	PLA-8	PLA-10
PLA/phr	100	100	100	100	100	100
BEP/phr	0	2	4	6	8	10
HMV-8CA/phr	0	0.06	0.12	0.18	0.24	0.3

**Table 2 polymers-11-01413-t002:** Isothermal crystallization kinetic parameters of various PLA/BEP composites.

Sample Numbers	*t*_1/2_/min	*n*	*K*	*T*_m_/°C	Δ*H*_m_/J·g^−1^	Crystallinity/%	*t*_m_/min
PLA	12.07	2.69	0.047	165.4	28.41	30.55	16
PLA-4	10.22	2.81	0.051	159.7, 167.0	40.67	45.48	14
PLA-8	7.11	2.89	0.075	160.3, 167.3	39.62	46.01	10

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
