# Peer review of "Toughened Poly(lactic acid)/BEP Composites with Good Biodegradability and Cytocompatibility"

_polymers, 2019, doi:10.3390/polym11091413_

Round 1
Reviewer 1 Report
The author talks about an intriguing result about Toughened poly(lactic acid)/BEP composites with good biodegradability and cytocompatibility. The experiments are generally topic to be interesting. However, there are some questions listed as follow.
After careful review, we believe that there is no good biodegradability and cytocompatibility in the data presented in this paper. Suggested topics and results need to be revised. PLA and BEP are both aliphatic, also should be partially compatible, so the notched impact strength increases reasonably, but why the elongation at break elongation increases. (p.5) SEM micrographs of the biodegradation of neat PLA and PLA/BEP composites should also be provided. Why does Tc recrystallization produce double peaks, which is related to the incompatibility between PLA and BEP?(p.6)
5. It is found that a4-a6 spherulites are more obvious, and b1-b3 is less significant. (p.8)
Author Response
Dear Reviewer:
We appreciate you very much for your positive and constructive comments and suggestions on our manuscript entitled “Toughened poly(lactic acid)/BEP composites with good biodegradability and cytocompatibility”. (ID: polymers-571334).
On the basis of your recommendation and comments, we have revised the manuscript, and the followings are our responses.
Point 1: After careful review, we believe that there is no good biodegradability and cytocompatibility in the data presented in this paper. Suggested topics and results need to be revised.
Response 1:
After comprehensively studying the PLA/BEP composites, We concluded that the PLA/BEP composites has good biodegradability and cytocompatibility, and the reasons are as follows.
In section 3.3, seen from Figure 9 and Figure 10, mass loss ratios of the PLA/BEP composites degraded in both soil and lipase solution were higher than those of neat PLA. Therefore, the PLA/BEP composites has better biodegradability than neat PLA. (It is well known that the PLA has been widely applied due to its good biodegradability.)
In Section 3.4, seen from Figure 12, the cell viability values measured from the extracts of PLA/BEP composites are higher than those measured from the extracts of neat PLA, and the cell viability values were higher than 75%, while the cell viability standard of ISO 10993-5: 2009 is 70 %. Seen from Figure 13, the extracts of PLA/BEP composites do not bring considerable changes in morphology of the cells such as rounding or shrinking. Therefore, we can conclude that the PLA/BEP composites has good cytocompatibility.
Point 2: PLA and BEP are both aliphatic, also should be partially compatible, so the notched impact strength increases reasonably, but why the elongation at break elongation increases.
Response 2:
The reason why the elongation at break of the PLA/BEP composites increase too is explained in Section 3.1. “These phenomena can be explained by the multiple crazing theory [36]. On the one hand, the well dispersed BEPs in PLA matrix has the effect on stress concentration during the tensile process, inducing the initiation and extension of crazing in PLA matrix. Both of the crazing initiation and the crazing extension can consume the energy. Accordingly, the polymer absorbs a high amount of energy and avoids a highly localized strain process, which lead to the higher elongation at break and impact strength[47][48]. On the other hand, the crazing extension will terminate when they meet with some other BEPs, thereby preventing the generation of cracking in PLA matrix and increasing the toughness of the PLA/BEP composites[49].”
Point 3: SEM micrographs of the biodegradation of neat PLA and PLA/BEP composites should also be provided.
Response 3:
Seen from Figure 8, surface morphology and the mildew growth of both neat PLA and PLA/BEP composites degraded in soil for 4 months can be seen clearly by the stereo microscopy images with low magnification. The surface of neat PLA after degradation remained smooth and clean, while surface of the PLA/BEP composites is rougher and extensive mildew was observed on the surfaces, which suffices to prove the differences between neat PLA surface morphology and the PLA/BEP composites surface morphology. Therefore, it is not necessary to use the SEM micrographes with high magnification to evalaute the surface morphology of neat PLA and PLA/BEP composite.
Point 4: Why does Tc recrystallization produce double peaks, which is related to the incompatibility between PLA and BEP?
Response 4:
The double melting peaks appeared in Figure 5 is not related to the incompatibility between PLA and BEP.
During the isothermal crystallization at 115 °C, the PLA/BEP composites produce two crystal forms, α and α' crystal forms, thus leading to two melting temperatures according to the two crystal forms. And the XRD results (Figure 6) also proves above two crystal forms.
Point 5: It is found that a4-a6 spherulites are more obvious, and b1-b3 is less significant.
Response 5:
Figure 7 has been revised to have equal panels for both neat PLA and PLA/BEP composites, and the a4, a5 and a6 have been removed.
Line 257 to line 269 in revised manuscript explained the differences between spherulites morphology of neat PLA and spherulites morphology of PLA/BEP composites.
The BEP in PLA matrix acts as the nucleating agent to increase the number of crystal nucleus, crystal forms and the crystallization rate during the PLA crystallization, and the spherulites overlap with each other, so that the shape of spherulites is not obvious. In contrast, the number of crystal nucleus, crystal forms and crystal rate of neat PLA is less than that of the PLA/BEP composites, the individual spherulites (for example, Figure a3 and the removed a4-a6) seem to be obvious.
Again, the authors appreciate your time, recommendation, and construction comments.
Thank you so much.
Sincerely yours
Qingguo Wang

Reviewer 2 Report
This manuscript presents the evaluation of PLA/BEP composite mechanical properties and biocompatibility. The study is thoroughly done and the results are interesting. The composite materials appear to have promise for further industrial and biomedical applications.
Comments
There is limited introduction and discussion in the manuscript. If the authors elaborated on their presented information in the introduction it would improve the manuscript. Likewise, if they better placed their results with the literature with the discussion, it would improve the manuscript.
The sample numbers should be provided in the methods section.
Error bars must be included for the cell viability data in Figure 13.
Figure 8 should be revised to have equal panels for both samples. Showing the extended time is not critical for the PLA samples, in my opinion.
Including Figure 1 in the introduction is not appropriate, in my opinion. It could be better added to the results section or removed.
Minor Comments
Please define the term phr.
Author Response
Dear Reviewer:
We appreciate you very much for your positive and constructive comments and suggestions on our manuscript entitled “Toughened poly(lactic acid)/BEP composites with good biodegradability and cytocompatibility”. (ID: polymers-571334).
On the basis of your recommendation and comments, we have revised the manuscript, and the followings are our responses.
Point 1: There is limited introduction and discussion in the manuscript. If the authors elaborated on their presented information in the introduction it would improve the manuscript. Likewise, if they better placed their results with the literature with the discussion, it would improve the manuscript.
Response 1:
As suggested by the reviewer, we have revised the Introduction section and Results and discussion section.
Point 2: The sample numbers should be provided in the methods section.
Response 2:
The sample numbers was defined in Table 1 (Recipes of various PLA/BEP composites) in Section 2.2 (Preparation of the PLA/BEP composites).
Point 3: Error bars must be included for the cell viability data in Figure 13.
Response 3:
As suggested by the reviewer, we have added the error bars in the Figure in revised manuscript.
Point 4: Figure 8 should be revised to have equal panels for both samples. Showing the extended time is not critical for the PLA samples, in my opinion.
Response 4:
As suggested by the reviewer, the Figure has been revised to have equal panels for both neat PLA and PLA/BEP composites, and the a4, a5 and a6 have been removed.
Point 5: Including Figure 1 in the introduction is not appropriate, in my opinion. It could be better added to the results section or removed.
Response 5:
As suggested by the reviewer, we have removed the Figure 1 in revised manuscript.
Point 6: Please define the term phr.
Response 6:
The “phr” is the abbreviation of “parts per hundred of rubber by weight”. As suggested by the reviewer, we defined the “phr” in a bracket after its full name for its first appearance in Abstract.
Again, the authors appreciate your time, recommendation, and construction comments.
Thank you so much.
Sincerely yours
Qingguo Wang

Round 2
Reviewer 1 Report
The authors answered the questions.